# Comparison and Characterization of Prenatal Nutrition Counseling among Large-for-Gestational Age Deliveries by Pre-Pregnancy BMI

**DOI:** 10.3390/nu11123018

**Published:** 2019-12-10

**Authors:** Kiley B. Vander Wyst, Guadalupe Quintana, James Balducci, Corrie M. Whisner

**Affiliations:** 1College of Health Solutions, Arizona State University, Phoenix, AZ 85007, USA; kbvanderwyst@asu.edu; 2Maricopa Integrated Health System, Phoenix, AZ 85008, USA; gquint28@gmail.com (G.Q.); James_Balducci@dmgaz.org (J.B.)

**Keywords:** nutrition, pregnancy, obesity, infant, fetal growth

## Abstract

It is recommended that prenatal care include nutrition counseling; however, <70% of women report receipt of nutrition counseling during pregnancy. In this study, we aimed to characterize prenatal nutrition counseling (PNC) among large-for-gestational age deliveries at a low-income and minority-serving hospital by performing a retrospective chart review of infants with a birth weight > 4000 g. Of the 2380 deliveries, 165 met the inclusion criteria. Demographics, PNC receipt, and pregnancy outcomes were compared among normal-weight (NW; BMI: 18.5–24.9 kg/m^2^, 19%, *n* = 31), overweight (OW; BMI: 25–29.9 kg/m^2^, 29%, *n* = 48), and obese (OB; BMI > 30 kg/m^2^, 52%, *n* = 86) women. The majority (78%, *n* = 129) of women were Hispanic White with a mean age of 30.4 ± 5.7 yrs and gestational weight gain of 12.1 ± 5.8 kgs. A total of 62% (*n* = 103) of women received PNC. A total of 57% gained above the Institute of Medicine (IOM) recommendations (*n* = 94). OB women were 2.6 and 2.1 times more likely to receive PNC than OW (95% CI: 1.1–2.0) and NW (95% CI: 0.9–1.9) women, respectively. Women who gained within the IOM recommendations for their pre-pregnancy body mass index (BMI) were 50% less likely to receive PNC than women who gained above the IOM recommendations for their pre-pregnancy weight (χ = 4.45, *p* = 0.035; OR = 0.48, CI: 0.24 to 0.95). Infant birthweight was significantly higher among women who received PNC (4314 ± 285 *vs.* 4197 ± 175 g, *p* = 0.004). These data suggest that PNC was directed toward women who enter pregnancy in the obese weight category and/or gain excessively across gestation. Future studies should provide PNC to all women to evaluate whether it reduces the risk of delivering large-for-gestational age deliveries across all maternal weight categories. Additionally, more work is needed to identify the types of PNC that are most effective for this high-risk population.

## 1. Introduction

Initiation of early and regular prenatal care is strongly associated with healthier infants and better pregnancy outcomes [1,2]. However, 47–72% of obese pregnant women [3] experience excessive gestational weight gain (EGWG), which is weight gain above clinical recommendations. EGWG coupled with pre-existing overweight or obesity increases the risk for maternal health complications such as miscarriage, gestational diabetes, and caesarean delivery, [4,5,6,7], and negative neonatal outcomes such as large-for-gestational age (LGA) infants [5], infant mortality [8], and childhood overweight or obesity [6]. Despite the high proportion of women beginning pregnancy overweight or obese, pregnant women do not consistently receive healthy diet and lifestyle counseling during prenatal care visits.

It is recommended that all pregnant women receive prenatal nutrition counseling (PNC) with referral to a registered dietician for an individualized assessment and dietary plan to optimize nutrition during pregnancy [1,2]. However, due to lack of reimbursement for these services and priority given to high-risk pregnancies, the registered dietician seldom provides PNC to otherwise healthy pregnant women [9]. Therefore, the delivery of PNC becomes the responsibility of the healthcare team, which varies by healthcare provider [10,11], with 19% and 26% of obstetricians and general practitioners, respectively, providing counseling [11]. Other studies report that ~69% of women received PNC from their healthcare provider [12,13] and that there was no difference in PNC by pre-pregnancy body mass index (BMI) [13]. Despite this, providers who are more knowledgeable about gestational weight gain (GWG) recommendations are more inclined to provide PNC [11]. Receipt of PNC has generally focused on vitamins and foods to avoid (i.e., lunch meat, high-mercury fish, etc.) and less on a well-balanced, healthy diet. Lack of PNC by healthcare providers results in women utilizing outside sources such as pregnancy books, magazines, and the internet, which might not contain the most reliable information [14]. There are a multitude of barriers to healthcare providers providing PNC that have been identified ranging from lack of time to inadequate training [14]. Not surprisingly, PNC is limited in scope [15] with varying frequency and quality indicating a missed educational opportunity for early counseling and intervention. 

There are several studies that have evaluated the impact of diet and lifestyle education among pregnant women. The LIMIT study found that overweight and obese women who received the lifestyle intervention improved their diet quality and activity level [16] but did not experience improved GWG [17]. This is similar to other studies that found no effect on total GWG [18] or adherence to GWG recommendations [19]. Other studies report that PNC effectively reduced total GWG [20,21,22] and postpartum weight retention [21] among normal-weight women. Although lifestyle interventions have not been effective at reducing EGWG [22,23] among overweight and obese women, it has resulted in reductions in birthweight and postpartum weight retention among overweight women [19]. These inconsistent findings are primarily due to the large methodological variability in dietary interventions aimed at controlling GWG, which makes consensus and development of evidence-based guidelines challenging. Therefore, it is important that we continue to evaluate the impact of PNC on high-risk maternal populations. 

Perinatal and prenatal nutrition counseling may play a vital role in maternal child health by increasing the number of women who enter pregnancy at a healthy BMI and gain appropriate gestational weight. The aim of this study was to compare pregnancy outcomes between women who did and did not receive prenatal nutrition counseling at a low-income and minority-serving hospital by pre-pregnancy BMI.

## 2. Materials and Methods

### 2.1. Methods 

A retrospective chart review was undertaken for all women who delivered a fetal macrosomic (birthweight > 4000 g) infant at Maricopa Medical Center (MMC) between 1 January 2015 and 31 December 2015. MMC is the only public teaching hospital in Arizona that primarily serves low-income, high-risk patients. Women who were 18 years of age or older were included in the study. Women were excluded if they had minimal prenatal care visits (<5 visits) or had late-initiation of prenatal care (i.e., beginning after 28 weeks gestation). Additionally, if women obtained any prenatal care at another institution and the records were not available they were excluded from this analysis. Figure 1 is a flow diagram of the number of charts screened to obtain the included participants.

The Information Technology Department at MMC conducted a data pull for all women who delivered during the study time period. Medical records were reviewed for inclusion and exclusion criteria. Data were extracted and entered into a password-protected Microsoft Excel spreadsheet. All participants were assigned a unique study identification number that was used to link the data to the medical record number. The linking document was maintained by the principal investigator on a password-protected computer. To ensure consistent quality and accuracy, all data were reviewed and extracted by the same research staff member (KBV).

### 2.2. Data and Definitions

Information extracted from the medical records included the following: maternal age, race/ethnicity, type of insurance, gravidity, parity, pre-pregnancy BMI, BMI at delivery, total gestational weight gain, mode of delivery, maternal comorbidities (diabetes, preeclampsia, hypertension, Group B Streptococcus status), pregnancy complications (i.e., hemorrhage, gestational age at first prenatal visit and delivery, maternal glucose readings during each trimester, last point of care glucose reading after delivery, type of nutrition counseling provided, healthcare provider(s) that provided PNC, exercise or physical activity counseling, ultrasound data (fetal biparietal diameter, femur length, abdominal circumference, head circumference), infant sex, birth weight, weight at discharge, Apgar scores, infant glucose readings, infant complications or comorbidities, and infant weight at 12 months.

Prenatal nutrition counseling was defined as any counseling provided by a member of the health care team during prenatal visits that included but was not limited to the following topics: diet, food, food groups, healthy foods and snacks, healthy eating, vitamins, specific macro- and micro-nutrients, fruits and vegetables, fiber, portion control and sizes, sodium, meals, hydration, caffeine, whole grains, meat and fish, and sugary beverages. The medical records were searched using these key terms and all prenatal visit records were reviewed for any documentation of PNC. The nutritional topics covered and documented in the medical record were recorded on the data collection sheet. The healthcare provider that documented and signed the note was recorded as the provider that delivered PNC. 

### 2.3. Statistical Analyses

Demographic characteristics and baseline factors were summarized using counts and percentages for categorical variables, and the mean and standard deviation or median and interquartile range for continuous measures for parametric and non-parametric data, respectively. Group differences between receipt and no receipt of PNC for demographic and anthropometric data were assessed using Chi-square or Fisher’s exact test statistics for categorical variables and independent t-tests for continuous variables. Odds ratios and relative risks were calculated to investigate differences in receipt of PNC based on pre-pregnancy BMI and Institute of Medicine (IOM) recommendation categories. Significance was considered when the 95% confidence interval excluded 1. Binary logistic regressions were performed to evaluate the effects of maternal age, infant sex, pre-pregnancy BMI category, and adherence to IOM recommendations on the likelihood that participants received PNC. One-way ANCOVA was performed to evaluate differences in GWG, infant anthropometric data at delivery, fetal biometry during second and third trimester, and infant weight at 12 months by receipt of PNC. The model for GWG included maternal age and pre-pregnancy BMI as covariates. The models for fetal biometry included infant sex and gestational age at ultrasound visit as covariates. The model for infant weight at 12 months included infant sex and maternal pre-pregnancy BMI as covariates. All statistical procedures were performed in SPSS Version 25 (SPSS, Chicago, IL, USA) and all statistical tests were two-sided with significance evaluated at the 5% level.

## 3. Results

### 3.1. Demographics

The mean age for the entire cohort was 30.4 ± 5.7 years with 78% (*n* = 129) of the women being Hispanic White and 56% (*n* = 93) stating their relationship status as single. The majority (52.1%, *n* = 86) of the women in the study were obese (BMI ≥ 30 kg/m^2^), followed by 29.1% (*n* = 48) as overweight (BMI: 25–29.9 kg/m^2^), and 18.8% (*n* = 31) being normal weight (BMI: 18.5–24.9 kg/m^2^). The average number of prenatal visits was 13.4 ± 5.1 visits. Women who received PNC had significantly more prenatal care visits, and higher pre-pregnancy and delivery BMI but earlier gestational age at delivery than women who did not receive PNC. Additionally, there was a greater proportion of cesarean deliveries among women who received PNC (47.1% vs. 24.2%, *p* = 0.004). There were no other statistically significant differences. Table 1 provides a summary of maternal demographic and anthropometric information by group.

There were no differences between groups (PNC vs. no PNC) for infant sex, weight at discharge, head circumference or length at delivery, or Apgar scores (Table 2). However, infants born to women who received PNC had a significantly higher mean birthweight (4313.8 ± 285.0 g) compared to infants of women who did not receive PNC (4196.9 ± 175.4 g; (t(163) = −3.26, *p* < 0.001). This remained statistically different after adjustment for pre-pregnancy BMI (PNC: 4305.9 ± 24.3 g vs. No PNC: 4209.8 ± 31.5 g; *p* = 0.018).

### 3.2. Prenatal Nutrition Counseling

The majority of women (62.4%, *n* = 103) received some kind of PNC during their prenatal care visits. PNC was primarily given by nurses (46%, *n* = 47), registered dietitians (32%, *n* = 33), and medical assistants (30%, *n* = 31). However, 25% (*n* = 26) of women received PNC from multiple healthcare providers. There were no statistically significant differences in the healthcare provider giving PNC by pre-pregnancy BMI, except for PNC from an RD. There was a greater proportion of obese women (31.4%, *n* = 27) who received PNC from an RD than overweight (8.8%, *n* = 3) or normal-weight (11.8%, *n* = 4) women (χ = 13.29, *p* = 0.001). The nutrition topics covered during PNC varied; however, the most frequently covered topics were general pregnancy diet and healthy foods and snacks. These were the top nutrition topics covered across all pre-pregnancy BMI categories. Figure 2 provides the nutrition topics covered during PNC. 

Evaluation of PNC by pre-pregnancy BMI revealed that obese women more frequently received PNC than overweight (72.1% vs. 50%, χ = 6.5, *p* = 0.1; OR = 2.58, 95% CI 1.24 to 5.40) and normal-weight (58.1% vs. 54.8%, χ = 3.09, *p* = 0.08; OR = 2.13, 95% CI 0.91 to 4.98) women. The majority (57%, *n* = 94) of women gained above the IOM recommendations for their pre-pregnancy BMI category. Figure 3 provides the proportion of women within each IOM weight gain recommendation category by receipt of PNC. 

Women who gained within the IOM recommendations where half as likely to receive PNC than women who gained above the IOM recommendations (71.3% vs. 54.2%, χ = 4.45, *p* = 0.035; OR = 0.48, 95% CI: 0.24 to 0.95). Table 3 provides the odds ratios, relative risks, and confidence intervals for receipt of PNC by BMI and IOM recommendation categories. No statistical differences between groups remained after investigating receipt of PNC and IOM recommendation adherence separately for each pre-pregnancy BMI category. 

A binary logistic regression was performed to ascertain the effects of maternal age, infant sex, pre-pregnancy BMI category, and adherence to IOM recommendations on the likelihood that participants received PNC. Pre-pregnancy obese BMI category (β = 0.90, *p* = 0.057) and gaining above (β = −0.75, *p* = 0.042) and below (β = 1.57, *p* = 0.026) the IOM recommendations were statistically significant predictors of PNC receipt. Infant sex (β = −0.14, *p* = 0.697) and maternal age (β = −0.01, *p* = 0.784) were not statistically significant predictors. The overall logistic regression model was statistically significant (χ^2^(6) = 15.85, *p* = 0.015). The model explained 13% (Nagelkerke R^2^) of the variance in receipt of PNC and correctly classified 67% of the participants. Women with an obese pre-pregnancy BMI were ~2.5 times more likely to receive PNC than women with a normal weight pre-pregnancy BMI after controlling for infant sex, maternal age, and adherence to IOM recommendations (OR = 2.448, 95% CI: 1.202, 19.326). Similarly, women who gained above the IOM recommendations were 2.1 times more likely to receive PNC when compared to women who gained within IOM recommendations after controlling for the same covariates (OR = 2.108, 95% CI 1.025, 4.329). 

### 3.3. Maternal and Fetal Health Outcomes

An ANCOVA revealed that there were no differences in gestational weight gain among women who did and did not receive PNC (12.9 vs. 12.7 kg, F(1) = 0.009, *p* = 0.925) after controlling for pre-pregnancy BMI, delivery mode, race/ethnicity, and parity. Evaluation of the fetal growth parameters during the second trimester by receipt of PNC revealed statistically significant differences for head circumference during the second trimester ultrasound. The fetuses of the women that did not receive PNC had significantly larger head circumferences than the women that received PNC (189.2 ± 1.1 vs. 186.9 ± 0.9 mm, F(1,129) = 5.29, *p* = 0.023) after controlling for infant sex and gestational age at the time of ultrasound. There were no other statistically significant differences for second- or third-trimester ultrasound growth parameters by receipt of PNC. Table 4 provides a summary of fetal growth parameters during the second and third trimesters by PNC receipt. There was no difference in infant weight at 12 months between women who received PNC compared to those who did not (10.7 ± 0.2 vs. 10.8 ± 0.3 kgs, F(1,87) = 0.00, *p* = 0.991).

## 4. Discussion

The present study evaluated the odds of pregnant women receiving nutrition counseling during prenatal care visits and the relationship nutrition counseling had with maternal, fetal, and infant health outcomes. Our primary findings indicate that receipt of PNC was greatest among obese women and women who gained weight above of the IOM recommendations; however, receipt of PNC was not associated with decreased GWG, differences in fetal biometry measures, or reduction in infant weight at 12 months. 

The current study found that women with an obese pre-pregnancy BMI were much more likely to receive some kind of nutritional counseling during prenatal visits when compared to their normal weight or overweight counterparts. Previous studies have found differences in nutrition counseling among various healthcare disciplines [10,11], but there are no studies evaluating the differences in nutrition counseling by maternal pre-pregnancy BMI categories despite evidence indicating a negative relationship between pregravid BMI and diet quality [24]. We also found that a higher proportion of women who received PNC gained above or below IOM recommendations when compared to those who did not receive PNC. This is similar to previous research that found receipt of PNC does not increase the proportion of women gaining within the IOM recommendations [19]. Regardless, it is important for health care providers to counsel pregnant women on the proper nutrition needed during pregnancy. 

The current study found no difference in GWG among women who did and did not receive PNC. This is similar to other studies that have found no difference in GWG [25,26] or postpartum weight retention [27] among women who received a nutritional education intervention compared to standard-of-care prenatal visits. However, a systematic review and meta-analysis demonstrated an overall reduction in GWG of 0.45 kg among intervention studies that provided nutrition counseling to pregnant women [28], which is in contradiction to the current study. Similarly, a prospective controlled study that provided an individualized diet plan in accordance with the recommended dietary allowance for a pregnant population, which consisted of three meals and two snacks per day, found weight gain was significantly lower among the intervention group as compared to the control group [29]. The lack of difference in GWG in the current study may be due to the women receiving PNC already having a higher pre-pregnancy BMI. Comparison of this finding is difficult as the majority of previous literature reports are on intervention studies.

The secondary outcomes of interest were to investigate the differences in fetal biometry measures and infant weight at 12 months among women who did or did not receive PNC. The current study found that infants of women who received no PNC had significantly greater head circumference at the second trimester than women who received PNC. Currently, there is a paucity in the literature of studies evaluating the impact of prenatal nutrition counseling on fetal growth and development. The studies that have evaluated the impact of nutrition counseling on fetal or infant health outcomes have largely focused on increased [30,31] or decreased [19,28] birthweight. Two of the studies reported 40 g [31] and 69.8 g [30] increases in birthweight among high-risk, low-income women participating in nutrition counseling and supplementation programs. On the other hand, intensive nutrition counseling among overweight and obese women resulted in a mean difference in birthweight of 157.4 g [19]. Incidence of increased birthweight was confirmed by a recent systematic review that found nutrition education during pregnancy resulted in higher mean birthweights of 64.8 g and 151.7 g among high-income and low/middle income countries, respectively [28]. This is similar to the current study that found higher birthweights among women who received PNC. A recent publication found that prenatal nutrition education did not result in a decreased incidence of rapid infant weight gain in the first two years of life or elevated offspring BMI percentile at two years of age [27]. The results of the present study are similar as we found no differences in infant weight at 12 months among women by receipt of PNC. Despite these negligible findings, the present study adds to the limited body of evidence regarding the effects of PNC on fetal development and infant health.

There were several limitations of the current study that are worth mentioning. The study was a retrospective chart review which limited the eligible participants to those who had comprehensive documentation of the necessary variables in the electronic health record system. Prenatal nutrition counseling information was also limited to what was documented by the healthcare provider during clinic visits. Because of this, we were unable to collect more detailed information about the provided PNC such as duration and frequency. The majority of PNC was not provided at a separate visit with a registered dietitian but was rather incorporated into the prenatal visit, limiting the potential information that was covered. Despite this, a retrospective chart review is an efficient, cost-effective way to initially assess the relationship of interest prior to conducting a more rigorous study. The generalizability of the study is limited to the same or a similar, high-risk, low-income, minority population, which may not be representative of the general population. Additionally, we cannot adequately evaluate the impact of receipt of PNC as we did not assess dietary intake of the women. There are a multitude of factors that may have led these women to not translate nutrition recommendations into behavior change or dietary modification. Lastly, the current study did not collect time series data so we were unable to extrapolate a causal relationship between PNC and maternal and infant anthropometric outcomes. Despite these limitations, there were several strengths of the current study. First, the study population consisted primarily of Hispanic White women of low socio-economic status. Therefore, this study adds to the current limited body of literature assessing nutritional counseling among a high-risk, minority population. Lastly, this study adds to the limited data on whether prenatal nutrition counseling is provided by health care providers and how these efforts impact maternal child health outcomes.

## 5. Conclusions

In conclusion, the present study found clear differences in receipt of PNC among pregnant women who delivered LGA infants by pre-pregnancy BMI and adherence to IOM recommendations. Women with an obese pre-pregnancy BMI and those who gained more than the recommended weight were significantly more likely to receive PNC than women not receiving PNC. However, the proportion of women who gained within the IOM recommendations was similar between those who receive and did not receive PNC, suggesting that other factors may have influenced their achievement of weight gain recommendations. Despite these findings, more research needs to be done to determine the true impact that receipt of nutrition education during pregnancy has on maternal child health. Future research should focus on evaluation of frequency and timing of nutrition education during pregnancy in order to understand the relationships between nutrition counseling and gestational weight gain, fetal growth and development, and infant weight trajectories. 

## Figures and Tables

**Figure 1 nutrients-11-03018-f001:**
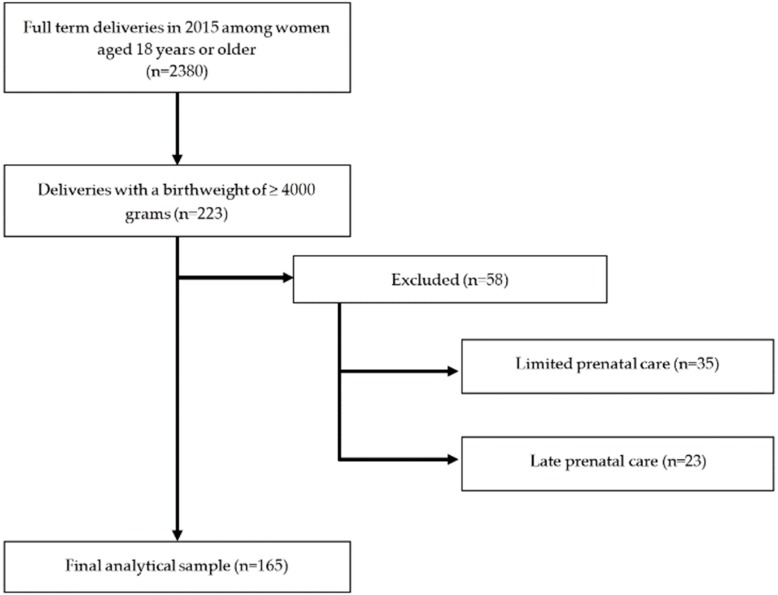
Flow diagram.

**Figure 2 nutrients-11-03018-f002:**
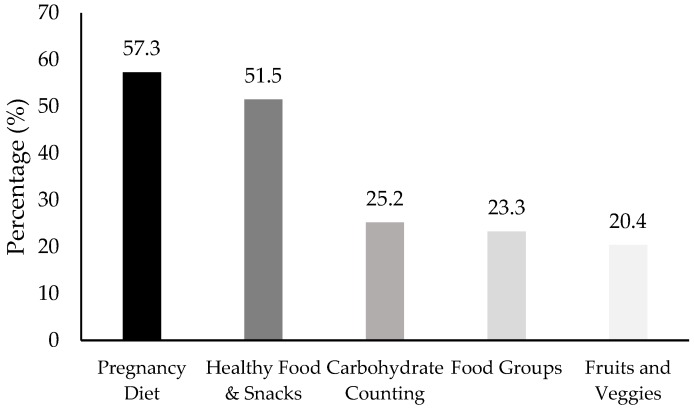
Top nutrition topics covered during prenatal visits and the percentage of women that received info on each individual topic.

**Figure 3 nutrients-11-03018-f003:**
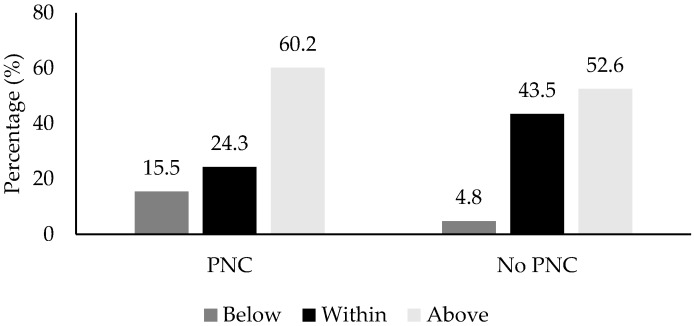
Adherence to IOM recommendations by receipt of PNC.

**Table 1 nutrients-11-03018-t001:** Maternal demographic and anthropometric information by receipt of prenatal nutrition counseling (PNC).

	PNC	No PNC	*p*-Value
Age, years (mean ± SD)	30.4 ± 5.4	30.4 ± 6.1	0.995
Race/Ethnicity, % (*n*)			0.518
Non-Hispanic White	7.8% (8)	4.8% (3)	
Hispanic White	73.8% (76)	85.5% (53)	
Black/African American	10.7% (11)	4.8% (3)	
Asian	4.9% (5)	3.2% (2)	
Other	2.9% (3)	1.6% (1)	
Marital Status, % (*n*)			0.528
Married	41.7% (43)	46.8% (29)	
Single	58.3% (60)	53.2% (33)	
Gestational Age (mean ± SD)			
First PNC	13.3 ± 5.9	13.4 ± 5.9	0.989
Delivery	39.8 ± 1.2	40.3 ± 0.9	0.003
# of Prenatal Visits (mean ± SD)	14.2 ± 6.0	12.1 ± 2.9	0.003
Gradivity (mean ± SD)	4.0 ± 2.2	3.8 ± 2.1	0.547
Parity (mean ± SD)	3.4 ± 1.8	3.3 ± 1.7	0.851
Delivery Mode, % (*n*)			0.004
Vaginal	52.9% (54)	75.8% (47)	
Cesarean Section	47.1% (48)	24.2% (15)	
Body Mass Index, kg/m^2^ (mean ± SD)			
Pre-pregnancy	31.4 ± 6.7	29.0 ± 5.3	0.014
Delivery	36.2 ± 6.5	33.6 ± 5.0	0.009
Pre-Pregnancy BMI, % (*n*)			0.025
Normal weight	16.5% (17)	22.6% (14)	
Overweight	23.3% (24)	38.7% (24)	
Obese	60.2% (62)	38.7% (24)	
IOM Recommendations, % (*n*) ^a^			0.012
Below	15.5% (16)	4.8% (3)	
Within	24.3% (25)	43.5 (27)	
Above	60.2% (62)	51.6 (32)	

Independent *t*-test was performed for continuous data and a Chi-square or Fisher’s exact test was performed for categorical data. Difference was deemed significant at the 0.05 level. Abbreviations: BMI = body mass index; PNC = prenatal nutrition counseling; IOM = Institute of Medicine. ^a^ IOM recommendations based on pre-pregnancy BMI for total gestational weight gain are: Underweight (BMI < 18.5 kg/m^2^) = 12.5–18.0 kg; normal weight (BMI between 18.5–24.9 kg/m^2^) = 11.5–16.0 kg; overweight (BMI between 25–29.9 kg/m^2^) = 7.0–11.5 kg; and obese (BMI > 30 kg/m^2^) = 5.0–9.0 kg.

**Table 2 nutrients-11-03018-t002:** Infant demographic and anthropometric data by receipt of PNC.

	PNC	No PNC	*p*-Value	Adjusted *p*-Value
Infant Sex, % (*n*)			0.981	
Male	61.9% (60)	61.7% (37)		
Female	38.1% (37)	38.3% (23)		
Birth Weight, g (mean ± SD)	4313.8 ± 285.0	4196.8 ± 175.4	0.004	0.018
Head Circumference, cm (mean ± SD)	36.1 ± 1.3	36.1 ± 1.3	0.808	0.995
Delivery Length, cm (mean ± SD)	53.4 ± 3.7	52.7 ± 4.3	0.323	0.313
5-min Apgar Score (mean ± SD)	7.9 ± 1.4	8.2 ± 1.2	0.153	0.220
10-min Apgar Score (mean ± SD)	8.9 ± 0.7	8.9 ± 0.6	0.850	0.734

Independent *t*-test was performed for continuous data and a Chi-square or Fisher’s exact test was performed for categorical data. One-way ANCOVA was performed to evaluate group differences after adjustment for pre-pregnancy BMI. These *p*-values are provided under the adjusted *p*-value column. Difference was deemed significant at the 0.05 level. Abbreviations: g = grams; in = inches; cm = centimeters; min = minute.

**Table 3 nutrients-11-03018-t003:** Odds Ratio, Relative Risks and Confidence Intervals for Receipt of PNC by BMI and Adherence to IOM Recommendations.

Variable	OR	95% CI	RR	95% CI
PNC by BMI (kg/m^2^) Category				
Obese vs. Overweight	2.58	1.24–5.40	1.44	1.06–1.97
Obese vs. Normal weight	2.13	0.91–4.98	1.32	0.93–1.86
Overweight vs. Normal weight	0.82	0.33–2.04	0.91	0.60–1.40
PNC by IOM Recommendations				
Above vs. Within	0.48	0.24–0.95	0.73	0.53–1.00
Above vs. Below	0.36	0.10–1.34	0.78	0.61–1.00
Within vs. Below	0.17	0.05–0.67	0.57	0.41–0.81

Relative risks were calculated for overall receipt of PNC by pre-pregnancy BMI category and adherence to IOM recommendations (reference categories are listed first). Abbreviations: BMI = body mass index; CI = confidence intervals; IOM = Institute of Medicine; OR = odds ratio; PNC = prenatal nutrition counseling; RR = relative risk.

**Table 4 nutrients-11-03018-t004:** Estimated marginal means and standard error for fetal growth. Parameters by receipt of prenatal nutrition counseling.

Fetal Biometry, Mean ± SD	PNC	No PNC	*p*-Value
Second Trimester			
Gestational Age, wks *	21.1 ± 0.3	21.1 ± 0.4	0.905
Estimated Fetal Weight, g	471.4 ± 8.3	467.0 ± 10.6	0.750
Biparietal Diameter, mm	49.3 ± 3.4	55.6 ± 4.4	0.259
Occipital Frontal Diameter, mm	67.6 ± 1.1	67.7 ± 1.4	0.955
Head Circumference, mm	186.0 ± 0.9	189.2 ± 1.1	0.023
Abdominal Circumference, mm	167.1 ± 1.7	168.4 ± 2.2	0.639
Femur Length, mm	35.8 ± 2.1	38.9 ± 2.7	0.354
Third Trimester			
Gestational Age, wks *	33.3 ± 0.5	33.7 ± 0.5	0.559
Estimated Fetal Weight, g	2789.4 ± 80.2	2815.6 ± 72.4	0.809
Biparietal Diameter, mm	84.9 ± 0.6	85.5 ± 0.7	0.54
Occipital Frontal Diameter, mm	112.0 ± 0.7	112.7 ± 0.8	0.541
Head Circumference, mm	315.5 ± 1.7	316.8 ± 2.9	0.617
Abdominal Circumference, mm	322.9 ± 3.4	322.3 ± 3.7	0.902
Femur Length, mm	66.3 ± 0.4	65.4 ± 0.5	0.15

A one-way ANCOVA was performed for all variables with gestational age at ultrasound visit and infant sex as a covariate except if otherwise noted. * Indicates one-way ANCOVA had maternal age as a covariate.

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
