# Peer review of "Comparison and Characterization of Prenatal Nutrition Counseling among Large-for-Gestational Age Deliveries by Pre-Pregnancy BMI"

_nutrients, 2019, doi:10.3390/nu11123018_

Round 1

Reviewer 1 Report

The authors conducted a retrospective study of in 165 pregnant women to characterize PNC among LGA deliveries followed in a low-income and minority-serving hospital. The authors found that a majority of women received PNC, which was more frequently observed in women with obesity. They also found that women who received PNC had larger babies, which is probably explained by obesity prior pregnancy. This paper is interesting; however, concerns are raised regarding methodology and results.

Abstract, lines 23-24 : “Women who gained within the IOM recommendations were half as likely to receive PNC than women who gained above the recommendations” : please rephrase that sentence, it is not clear.

Abstract, lines 20-21: Please separate those statements in two sentences. One for PNC, one for gestational weight gain.

Introduction: “However, 47-72% of obese pregnant women experience excessive gestational weight gain (EGWG),or gain above clinical recommendations’’. What is the difference between having EGWG and gain above clinical recommendations. I would say it is the same.

Introduction, lines 43-54: The authors should discuss the role of the dietician, who is the expert in providing PNC.

Intervention, objective : I’m not sure the authors had characterized prenatal nutrition, since limited information were reported in medical chart. The aim of the study was to compare pregnancy outcomes between women who received or not PNC.

Methods, lines 91-107: Please add a subsection for stats.

Methods, why the authors only looked at the year 2015? Why this year was chosen?

Methods, the authors should define what they consider as PNC. How was the data obtained? How was PNC delivered?  How the authors know about topics covered and who delivered PNC? A complete section should be added to describe that information, which woulb be consistent with results.

Methods, please list all variables that were collected in medical records. Add information on ultrasound.

Results – Table 1: I suggest the authors to adjust there results for pre-pregnancy BMI: a column could be added with adjusted  P-Value. “However, infants born to women who 123 received PNC had a significantly higher mean birthweight (4313.8±285.0 kg) compared to infants of 124 women who did not receive PNC (4196.9±175.4 kg; (t(163)=-3.26, p<0.001) ”. Was that difference remained significant after adjustment for pre-pregnancy BMI.

Figure 3- It is unusual to present data in that order. Please present, below, within and above, in that order.

Discussion: “This retrospective chart review suggests that receipt of prenatal nutrition counseling may not impact health as much as previously reported”. I’m not sure the authors can conclude that, with that study design. To conclude that, the authors should have compared women receiving or not PNC within the same pre-pregnancy BMI category, however, the sample size maybe too small. Furthermore, what if those pregnant women did not receive PNC, do pregnancy outcomes would have been worse?

Discussion lines 208: A significant difference of 0,45kg may not be clinically significant…. Furthermore, I’m not sure the authors can compare there results to those studies, since all studies included in that review were intervention studies.

Discussion, lines 219+: Data that are obtained from the second trimester ultrasound should be interpreted with caution. Those are imperfected estimations. This should be discussed.

A last important point: Women who received PNC may not have translate those recommendations into their diet for multiple reasons. The only way to know that would have been to assess dietary intakes with appropriate tools, before and after PNC. Barriers and facilitators to apply changes in diet after PNC should be addressed in future studies.

Reviewer 2 Report

This article would be strengthened with a clearer description of how  PNC was noted.  Was there a difference in results when patients had more than one PNC?  It is possible that PNC was very brief in some instances especially when provided by a nonRD and this should be noted.  

Round 2

Reviewer 1 Report

The authors have satisfactorily responded to all my comments.